# ATTENTION GUIDED ALIGNMENT IN EFFICIENT VISION-LANGUAGE MODELS

## ABSTRACT

Large Vision-Language Models (VLMs) rely on effective multimodal alignment between pre-trained vision encoders and Large Language Models (LLMs) to integrate visual and textual information. This paper presents a comprehensive analysis of attention patterns in efficient VLMs, revealing that concatenation-based architectures frequently fail to distinguish between semantically matching and non-matching image-text pairs. This is a key factor for object hallucination in these models. To address this, we introduce Attention-Guided Efficient Vision-Language Models (AGE-VLM)—a novel framework that enhances visual grounding through interleaved cross-attention layers to instill vision capabilities in pretrained small language models. This enforces in VLM the ability "look" at the correct image regions by leveraging spatial knowledge distilled from the Segment Anything Model (SAM), significantly reducing hallucination. We validate our approach across different vision-centric benchmarks where our method is better or comparable to prior work on efficient VLMs. Our findings provide valuable insights for future research aimed at achieving enhanced visual and linguistic understanding in VLMs.

## 1 INTRODUCTION

Large Vision-Language Models (VLMs) Alayrac et al. (2022); Liu et al. (2023); Radford et al. (2021); Tong et al. (2024a); Wang et al. (2024); Zhu et al. (2023) leverage the capabilities of pre-existing Large Language Models (LLMs) Achiam et al. (2023); Chowdhery et al. (2023); Grattafiori et al. (2024) to address complex tasks. Although LLMs excel in text-only domains such as natural language understanding Grattafiori et al. (2024), mathematics Cobbe et al. (2021), and coding Le et al. (2022) by following task-specific instructions, VLMs extend these abilities to the multimodal realm. This enables them to perform tasks like image understanding, captioning, object localization, and multi-turn visual question answering. Architecturally, VLMs typically consist of three key components: a vision encoder to process visual input, an adapter to map visual representations into the language model's token space, and a decoder-only LLM that processes these combined representations. The fusion of visual and textual information is commonly achieved either by concatenating visual tokens with text tokens for processing by self-attention layers or by interleaving visual tokens using dedicated cross-attention layers within the LLM Alayrac et al. (2022); Grattafiori et al. (2024).

Recent research Rahmanzadehgervi et al. (2024); Tong et al. (2024a;b) has shown a significant challenge in VLMs: a tendency to ignore visual modality representations when performing vision-language tasks. These models may produce answers, whether correct or incorrect, relying solely on textual instructions and associated questions, thereby ignoring crucial visual information. Tong et al. Tong et al. (2024a) observe that the performance gap with and without visual information is less than a 5% on multiple benchmarks including MMMU Yue et al. (2024), MathVista Lu et al. (2023), and AI2D Hiippala et al. (2021). To mitigate this, various efforts have focused on enhancing visual capabilities, including curating vision-centric datasets Tong et al. (2024a) and improving vision-text alignment through auxiliary mechanisms such as specialized projection mechanisms Masry et al. (2025). However, these limitations often persist even in efficient frameworks designed for smaller-scale models with fewer parameters. Despite such enhancements, many multimodal approaches still struggle with effective visual information processing, frequently exhibiting issues like object hallucination Guan et al. (2024) (as illustrated in Fig. 1).

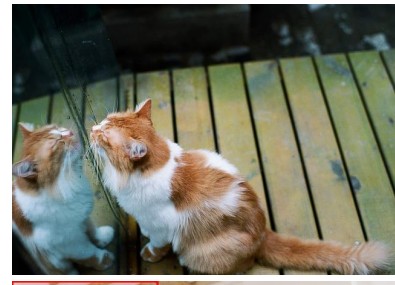

Prompt: How many cats are in the image? Select from the following choices. (A) 2 (B) 1 (C) 4 (D) 3 (E) 0

🎯 Groundtruth: (B)

❌ MobileVLM-V2: 3

✅ **Ours: The correct answer is (B) 1. The image shows only one cat, which is the orange and white cat looking at its reflection in the glass. There are no other cats present in the image.**

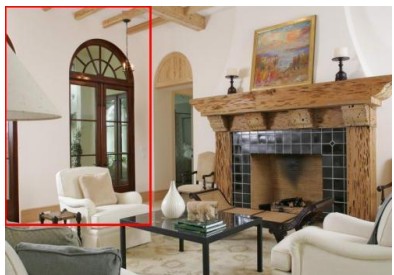

Prompt: Considering the relative positions of the wall (annotated by the red box) and the fireplace in the image provided, where is the wall (annotated by the red box) located with respect to the fireplace? Select from the following choices. (A) left (B) right

🎯 Groundtruth: (A)

❌ MobileVLM-V2: The image shows a living room with a fireplace, a couch, and a chair. The wall (annotated by the red box) is located to the left of the fireplace. The fireplace is in the middle of the room. The couch is to the right of the fireplace. The chair is to the left of the couch. The relative positions of the objects are as described. The answer is B.

✅ **Ours: The wall (annotated by the red box) is located to the left of the fireplace.**

Figure 1: **Object hallucination and mitgation in efficient vision language models.** We show that prior work on efficient VLMs fails to localize (top) and correctly count the number of objects (cat) in images. Our attention guided efficient VLMs approach equipped with the knowledge distillation from Segment Anything Model in the cross-attention weights can effectively generate correct responses with explanation grounded in the visual domain.

Many efficient VLMs Ge et al. (2024); Vasu et al. (2024) employ convolutional vision encoders like ConvNeXt Liu et al. (2022). To enable vision-language capabilities, ConvNeXt is trained contrastively with CLIP Ge et al. (2024). However, this approach leads to lack of fine-grained spatial grounding in the visual features. During standard VLM training, given the LLM's strong language prior and the use of next-token prediction with cross-entropy loss, the model fails to recover the fine-grained visual information. While larger models Lin et al. (2023); Wang et al. (2024) can integrate features from multiple encoders (e.g., DINO Oquab et al. (2024) , ViTDosovitskiy et al. (2021)) to improve grounding, this is unfeasible for resource-constrained VLMs.

To endow the efficient models with spatial grounding of the vision features and to mitigate object hallucination, we propose a novel framework called *Attention-Guided Efficient VLM* (AGE-VLM). Our approach modifies a standard LLM by interleaving cross-attention layers with its existing self-attention layers. The core idea is to distill spatial knowledge from the Segment Anything Model (SAM) Kirillov et al. (2023) directly into these cross-attention mechanisms. This is achieved by optimizing the cross-attention weights to align with segmentation masks generated by SAM for relevant text queries. Consequently, the VLM learns to "look" at the correct regions of interest when processing multimodal inputs. A key advantage is the data efficiency of this distillation process, enabling enhanced grounding with limited training examples. We make following contributions:

- We analyze the vision and text features in the hidden states of the efficient VLMs and uncover their limitations in disambiguating the semantics between similar and dissimilar image-text pairs to uncover limitations in vision-centric tasks including object hallucination.

- To endow relatively small LLMs (1B parameter models) with vision capabilities in an efficient manner, we propose a new efficient multimodal framework with cross-attention layers which leverage attention-guidance from segmentation model (SAM).

- To distill knowledge from the SAM model, we introduce a four stage training paradigm which seamlessly integrates the vision features with the pretrained LLM without effecting the language capability of the underlying model. Our efficient AGE-VLM with guidance loss only during pretraining stage outperforms prior art across vision-centric tasks. We observe notably high performance gains on vision benchmarks that require fine-grained image understanding such as POPE and OCR-bench.

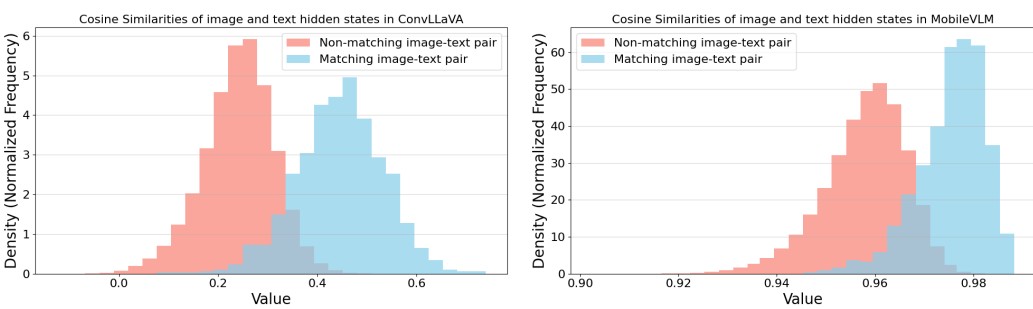

(a) Similarity for ConvLLaVA Ge et al. (2024)  (b) Similarity for MobileVLM-v2 Wu et al. (2024).

Figure 2: **Similarity analysis.** Cosine similarly between the hidden states of the images and text tokens of the last self-attention layer in existing efficient architectures. The similarities overlap for non-matching pairs indicating a gap in the alignment of visual signal with text.

## 2   RELATED WORK

**Efficient VLMs.** Vision–language models (VLMs) combine a visual encoder with a large language model to support multi-modal reasoning. Recent research has explored making VLMs more efficient and compact without sacrificing performance by using compressed image embedding and a smaller-sized language model. ConvLLaVA Ge et al. (2024) swaps the standard ViTDosovitskiy et al. (2021) for a ConvNeXtLiu et al. (2022) encoder, cutting the number of visual-token for high-resolution images. FastVLM Vasu et al. (2024) introduces a hybrid vision encoder that yields far fewer tokens and achieves a better speed-accuracy trade-off. MobileVLM Wu et al. (2024) reaches real-time speed on edge devices through extensive ablation of design choices. VL-Mamba Qiao et al. (2024) replaces the Transformer with linear-time state-space (SSM) layers, delivering near-linear scaling in sequence length while retaining competitive accuracy. Mini-GeminiLi et al. (2024) adopts a dual-encoder scheme (low-res ViT plus high-res ConvNeXt); the visual-token budget stays fixed, and high-resolution details are injected only when needed. Finally, TinyLLaVA Zhou et al. (2024), AppVLM Papoudakis et al. (2025), and VILA Lin et al. (2024) report further gains from better training recipes, thorough dataset curation, and deep understanding of pre-training.

**Attention in VLMs.** Most modern VLMs fuse vision and language information through carefully designed attention layers for comprehensive reasoning Bhattacharyya et al. (2024); Cobbe et al. (2021). FlamingoAlayrac et al. (2022) encodes images with a Perceiver module and feeds them into the language model via gated cross-attention. BLIP-2Li et al. (2023a) uses a Q-Former that queries image features and hands a compact token set to the LLM, injecting visual clues at multiple points to the language model with relatively few new parameters. Recent studies also highlight limitations in existing attention patterns and propose remedies. Zhang et al. Zhang et al. (2025) shows that performance falls sharply when the target object is small; a training-free, attention-guided cropping strategy recovers most of the lost accuracy. LRRBhattacharyya et al. (2024) interleaves top-down cross-attention blocks amid the LLM's self-attention, grounding generation in fine-grained video frames. Kang et al. Kang et al. (2025) observes biased attention toward irrelevant visual tokens and introduces visual-attention-sink suppression to redistribute focus and boost accuracy.

**Hallucination in VLMs.** Hallucination is well-known chronic problem in LVM. Misalignment between visual evidence and textual generation, especially in cluttered scenes, often drives such errors. Several benchmarks reveal that some VLMs perform similarly with or without visual input, implying that the language model may ignore image cuesGoyal et al. (2017); Kumar et al. (2024); Li et al. (2023c); Zohar et al. (2024); Zhang et al. (2024) To enhanced alignment, EMMAGhazanfari et al. (2024) balances structural and hierarchical representations, reducing hallucinated objects and sharpening visual grounding. Modular attribution studies find that multi-head attention poses higher hallucination risk than MLP blocks; disabling "hallucination heads" yields simple yet effective mitigation Yang et al. (2025). New evaluation suites now include explicit hallucination tests Guan et al. (2024); Li et al. (2023b); Tong et al. (2024a). For instance, the Multi-Object Hallucination Chen et al. (2024) dataset probes scenarios where models overlook vivid visual clues; its carefully curated corner cases trace errors to language bias and skewed object distributions.

## 3 Vision in Vision-Language Models

In this section, we will first investigate the underlying cause of object hallucination and limitations in processing the visual information in efficient VLMs. Based on our findings, we then propose a framework to mitigate this.

### 3.1 Attention analysis in VLMs

To understand the underlying causes of object hallucination and the tendency of VLMs–which are built on existing LLM backbones–to underutilize visual features, we analyze the semantic alignment between hidden states derived from their image and text modalities. Specifically, we compute the cosine similarity between the final hidden states of image tokens (or their projected representations) and text tokens, as visualized in Fig. 2. Our analysis considers two distinct concatenation-based models: ConvLLaVA, which pairs a convolutional vision backbone with a LLaMA-7B language model, and MobileVLM-v2, which utilizes a CLIP ViT-L/14 vision encoder with a LLaMA-1.4B language model.

For both ConvLLaVA and MobileVLM-v2, a critical observation is the significant overlap in similarity score distributions between matching (ground-truth image-text pairs) and non-matching (randomly paired images and texts from the batch/dataset) examples. This suggests these architectures systematically struggle to distinguish semantically coherent visual-textual pairs from incoherent ones using their hidden representations.

For ConvLLaVA (Fig. 2a), although the similarity scores for non-matching pairs are appropriately skewed towards lower values ($\sim 0.2-0.3$)–demonstrating some discriminative ability–the distribution for matching pairs is disappointingly centered around a modest $\sim 0.5$. Ideally, correctly matched pairs should exhibit a distribution strongly skewed towards higher scores (e.g., $> 0.8$), signifying robust alignment between visual concepts and textual descriptions. This current observation implies that even when an image and text are semantically related, their respective hidden states are not achieving the desired close alignment in the shared embedding space.

MobileVLM-V2 (Fig. 2b) exhibits indiscriminately high similarity scores for both matching and non-matching pairs, with both distributions peaking at very high values (e.g., $\sim 0.96 - 0.98$). This consistent high similarity, irrespective of actual image-text semantic relevance, suggests a critical limitation in its ability to capture meaningful underlying multimodal semantic information.

This behavior is a strong indicator for object hallucination in VLMs and the dependence of the generation process on the strong language priors. Indeed, if non-matching pairs consistently achieve high similarity scores, it implies that visual features are failing to sufficiently constrain the LLM. Consequently, generation becomes unanchored from the visual input, driven instead by the LLM's internal biases or textual context, which leads to both object hallucination and a tendency to disregard specific visual details.

To mitigate object hallucination in efficient VLMs, our work introduces cross-attention layers whose attention weights are distilled from the Segmentation Anything Model (SAM), thereby better grounding the pretrained LLM in visual information.

### 3.2 Attention guided Efficient VLM Approach

We present AGE-VLM, an efficient multimodal model that seamlessly integrates visual features with a language model architecture. AGE-VLM employs a ConvNext vision encoder and the LLaMA-1B decoder-only language model. The vision features are modulated by text tokens through cross-attention layers which are explicilty guided by distilling knowledge from SAM as illustrated in Fig. 3.

#### 3.2.1 Efficient Vision-Language Architecture

**Efficient vision backbone.**   Similar to prior VLMs employing convolutional backbones, we utilize a ConvNeXT to extract visual features. Convolutional networks advantageously process higher-resolution images with fewer visual tokens compared to ViTs. Given an input image $I$ of spatial

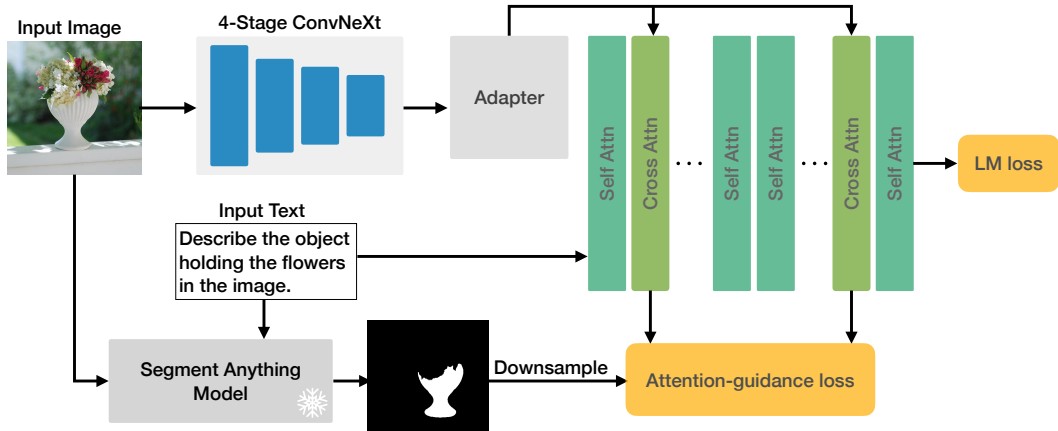

Figure 3: **Overall architecture of our attention-guided efficient vision language model.** During training, given the input image and the associated instruction, we perform knowledge distillation from SAM by explicitly aligning the language-conditioned masks with the cross-attention weights of our modified LLaMA-1B backbone.

resolution $H \times W$, the ConvNeXT backbone processes it through multiple convolutional stages[1]. We extract the spatial feature map $I'$ from the output of the fourth stage, which retains spatial information crucial for detailed visual understanding. This map $I'$ is then flattened and projected by two linear layers into a sequence of $h \times w$ visual tokens, each with dimension $d$ to match our language model's embedding dimension.

**Efficient LLaMA-1B backbone.** We employ LLaMA-1B as our language backbone, selected for its relatively small size, making it suitable for resource-constrained scenarios. The model processes tokenized text sequences. During training, most of LLaMA's parameters–specifically its self-attention and feed-forward network (FFN) weights–are kept frozen to preserve its powerful language priors and reduce training costs.

**Interleaved cross-attention layers.** To directly integrate visual information into the language model, we introduce cross-attention mechanisms within the LLaMA architecture. Instead of a simple prefix or concatenation approach, we interleave lightweight cross-attention modules into specific LLaMA decoder blocks. For LLaMA-1B, which has 16 decoder layers, these cross-attention modules are strategically inserted. A standard LLaMA decoder layer $i$ typically processes input hidden states $H_{i-1}$ as follows:

$$H_i = \text{SELFATTENTION}(\text{LAYERNORM}(H_{i-1})),$$
$$H_i = H_i + H_{i-1},$$
$$H_i = H_i + \text{MLP}(\text{LAYERNORM}(H_i)). \tag{1}$$

To retain the language capabilities of the pre-trained LLM, we modify select layers in LLaMA-1B–by inserting a cross-attention module after the standard self-attention sub-layer. This inserted cross-attention module takes the output of the self-attention sub-layer, $H_i$, and the visual features $I'$ (extracted by the ConvNeXT encoder and transformed by an adapter to match the LLM's hidden state dimension) as input. Within this cross-attention module, the hidden states from the self-attention sub-layer serve as queries $Q = W_q(H_i)$, while the transformed visual features serve as keys $K = W_k(I')$ and values $V = W_v(I')$. The operation is then:

$$H_{\text{CA}} = \text{CROSSATTENTION}(Q, K, V),$$
$$H_{\text{CA}} = H_i + H_{\text{CA}},$$
$$H_{\text{CA}} = H_{\text{CA}} + \text{MLP}(\text{LAYERNORM}(H_{\text{CA}})). \tag{2}$$

This interleaved structure allows the model to dynamically ground textual concepts in visual features at multiple semantic levels within the LLM. The weights $W_q, W_k, W_v$ and the parameters within

---

[1]Not to be confused with VLM training stages.

Table 1: Training stages of our Attention-guided Efficient Vision Language Model.

| Training Stage | Model | | | | Training Loss | |
|---|---|---|---|---|---|---|
| | Vision Encoder | Adapter | LLM(CA) | LLM(SA) | LM Loss | Guidance Loss |
| Stage 1 | ❄ | 🔥 | 🔥 | ❄ | ✓ | ✗ |
| Stage 2 | 🔥 | 🔥 | 🔥 | ❄ | ✓ | ✗ |
| Stage 3 | 🔥 | 🔥 | 🔥 | ❄ | ✓ | ✓ |
| Stage 4 (AGE-VLM) | 🔥 | 🔥 | 🔥 | 🔥 | ✓ | ✗ |
| Stage 4 (AGE-VLM-LM) | 🔥 | 🔥 | 🔥 | 🔥 | ✓ | ✓ |

the CrossAttention blocks with multi-head attention are trainable. In 4, we show the advantage of cross-attention layers for visual data. The time to generate first token, critical in practical deployment of VLMs increases exponetially with the increase in image resolution for self-attention mechanism while it remains constant for cross-attention based models.

### 3.2.2 SPATIAL GROUNDING DISTILLATION USING SAM

To account for the lack of spatial localization in VLMs optimized with the cross-entropy loss for next token prediction, we perform knowledge distillation from the Segment Anything Model in the cross-attention layers our model. For this, during pretraining stage, we take the 77K image-caption pairs corresponding to $\sim 10$ percent of the pretraining data of Cambrian 2.5M. Analogously, during fine-tuning, we take the 150K image-instruction or image-question pairs from the Cambrian 10M. Using these language queries, we obtain the language-grounded masks for the images using SAM.

Given the image $I$ of spatial resolution $H \times W$, and the text prompt query $t_q$ (ignoring the special tokens) we obtain the mask $M \in \{0,1\}^{H \times W}$. The mask is then downsampled to match the vision feature encoder's spatial resolution $h \times w$ yielding $M' \in \{0,1\}^{h \times w}$. Given the attention weights $A_l$, output of softmax in cross-attention layer corresponding to layer $l$, where $l \in l_1, \ldots, l_n$ and $n$ is the number of cross-attention layers, the attention weights for the query token $t_q$ are averaged across all the heads in the attention layer and are reshaped to $h \times w$ yielding $A_l^q$ . Consider an example of a text prompt with 10 tokens and 576 image tokens, the cross-attention layers with 32 heads would ouput the attention weights $A_l$ of size $32 \times 10 \times 576$. These weights are averaged along the first two dimensions providing 576 dimensional $A_l^q$. These attention weights are then normalized to obtain a attention distribution $P_l^q$, We perform distillation using the dice loss to localize the attention maps on the region represented by the mask,

$$\mathcal{L}_g = -\log \left[ \frac{2.\langle \text{vec}(M').\text{vec}(P_l^q) \rangle}{\sum_{i,j} M'_{i,j} + \sum_{i,j} P_l^q} \right] \tag{3}$$

Here, vec(.) flattens the input to a 1-d representation. The advantage of dice loss is that it directly optimizes for the overlap between predicted and ground truth masks accounting for sparse regions of interest which are otherwise difficult to optimize using binary cross-entropy loss. This loss is applied to all the cross-attention layers to modulate the visual features with text tokens.

The overall training objective is the sum of the standard $\mathcal{L}_{LM}$ – the causal language modeling (next-token prediction) loss computed using standard cross-entropy on the entire dataset and $\mathcal{L}_g$–the loss of distillation calculated on the subset with the SAM grounding masks. The two loss are trained with equal weights.

### 3.2.3 TRAINING STAGES

Our model training, detailed in Table 1, proceeds through four distinct stages. The first three stages comprise a comprehensive pre-training phase aimed at effectively aligning visual features with the textual representations of our lightweight 1B parameter LLM. A primary objective throughout this pre-training is to instill the LLM with visual capabilities while preserving its inherent language proficiency.

**Stage1: Initial Vision-Language Alignment.** The first stage establishes a foundational mapping between modalities by aligning visual features (processed through an adapter) with the LLM's textual representations. We achieve this alignment using newly integrated cross-attention layers, with training

Table 2: **Quantitative evaluation.** Comparison of our AGE-VLM with state-of-the-art efficient VLMs on vision-centric benchmarks.

| Method | HallusionBench | | | OCRBench | | CV-Bench | | RWQA | POPE |
|--------|------|------|------|--------------|-------------------|------|------|------|------|
| | aAcc | fAcc | qAcc | Scene Centric | Key Info. Extract. | 2D | 3D | | |
| CA-Baseline | 40.38 | 13.87 | 11.21 | 148.00 | 51.00 | 0.62 | 0.50 | 0.47 | 85.11 |
| ConvLLaVA | 24.71 | 8.96 | 4.84 | 117.00 | 26.00 | 0.59 | **0.57** | **0.51** | 77.76 |
| mobile-vlm-v2 | **44.37** | 14.45 | **11.65** | 101.00 | 2.00 | 0.31 | 0.40 | 0.28 | 84.30 |
| AGE-VLM | 43.85 | **15.32** | 11.21 | **149.00** | **59.00** | 0.61 | 0.52 | 0.48 | **87.34** |
| AGE-VLM-LM | 39.22 | 11.56 | 7.91 | 126.00 | 33.00 | **0.66** | 0.46 | **0.51** | 85.18 |

guided exclusively by the LLM's inherent language modeling objective (e.g., next-token prediction). This provides strong initial weights for the adapter and cross-attention modules, teaching them to map visual information into the LLM's embedding space. This methodology, utilizing image-caption pairs from the Cambrian 2.5M dataset, is analogous to the initial pre-training phase of standard VLMs.

**Stage 2: Vision Encoder Adaptation.** In the second stage, we unfreeze and fine-tune the final block of the ConvNeXT vision encoder, training it jointly with the adapter and cross-attention layers. This approach is motivated by prior work demonstrating that adapting pre-trained ConvNeXt models from their original resolution (e.g., $384 \times 384$) to higher resolutions (e.g., $768 \times 768$) enhances detailed visual understanding. Operating at this higher resolution, our ConvNeXT yields 576 visual tokens, comparable to a Vision Transformer (ViT) backbone at a $336 \times 336$ resolution. This highlights ConvNeXT's greater token efficiency compared to common ViT-based VLMs. The Cambrian 2.5M dataset continues to provide image-caption pairs for the LLM loss in this stage.

**Stage 3: Spatial Grounding via Knowledge Distillation and Alignment.** The third stage enhances visual grounding by incorporating knowledge distillation from the Segment Anything Model (SAM) to ensure generated responses are explicitly tied to relevant visual information. For approximately 10% of the Cambrian 2.5M image-text pairs, SAM generates segmentation masks for key entities or concepts relevant to the image-text context. We then optimize our model's cross-attention weights to align with these SAM-generated masks using the objective defined in Eq. 3. This encourages the cross-attention mechanism to focus on pertinent image regions during visual processing, thereby improving spatial grounding. The LLM loss is computed using the Cambrian 2.5M dataset.

**Stage 4: Visually Grounded Instruction Fine-tuning.** The final stage consists of end-to-end instruction fine-tuning for the entire model. We consider two variations for training. In the first setting, we follow Tong et al. (2024a) (AGE-VLM) and finetune the model without the attention-grounding loss. The key advantage of this is that the self-attention layers of the LLM are kept intact, allowing to efficiently integrate multimodal signal with the model retains its language capacity. In the second scenario (AGE-VLM-LM) visual grounding is maintained by concurrently applying the distillation loss (from Stage 3) and the primary LLM loss (next-token prediction for instruction following). For knowledge distillation, SAM is prompted with the instruction (typically a question) and its ground-truth answer. This guides SAM to generate segmentation masks for image regions most pertinent to that specific instruction-answer pair, and our model's attention is then distilled towards these masks. The LLM loss in this stage utilizes the full Cambrian 10M instruction-following dataset, while the grounding loss is applied to a 10% subset thereof, reinforcing the model's focus on relevant visual evidence.

## 4 EXPERIMENTS

To validate the effectiveness of our AGE-VLM in encoding and utilizing visual features to mitigate object hallucination, following Tong et al. (2024a) we perform extensive experiments on vision-centric tasks for objection hallucination evaluation on Visual Question Answering (VQA) with human edited images on HallusionBench Guan et al. (2024) and on the POPE Li et al. (2023b) dataset. Additionally, we include OCRBench Liu et al. (2024) for scene-centric text-VQA and for key information extraction from the receipt images. We further evaluate on CV-Bench and RealWorldQA x.ai (2024) to uncover multimodal capabilities in 2D tasks, i.e., spatial relationships or object count and for 3D tasks such as depth and relative distances.

Table 3: Qualitative comparison of our approach against Conv-LLaVA and MobileVLM-v2 on vision-centric task.

| | | | |
|---|---|---|---|
| Prompt | Is there a laptop in the image? Please answer yes or no. | Is there an apple in the image? Please answer yes or no. | Is there a cup in the image? Please answer yes or no. |
| Groundtruth | yes | yes | yes |
| Conv-LLaVA Ge et al. (2024) | 0 | 0 | no |
| MobileVLM-v2 Wu et al. (2024) | No, there is no laptop in the image. The image features a person typing on a keyboard. | No, there is no apple in the image. The image features a shelf with various items, including books, a teddy bear, and a Christmas ornament. | No, there is no cup in the image. The image only shows a plate of food, including a salad and a bowl of soup. |
| AGE-VLM | Yes, there is a laptop in the image. It is partially visible in the background, partially obscured by the hands of the person typing on the keyboard. The laptop is positioned to the left side of the image, and it appears to be a standard model with a white exterior and black keys. | Yes, there is an apple in the image. It is located on the top shelf, to the left of the teddy bear. The apple is red with a yellow stem and is placed among other items. | Yes, there is a cup in the image. It is located on the right side of the plate, partially obscured by the bread. The cup appears to be a clear glass, and it is filled with a transparent liquid, which could be water or another clear beverage. |

**Training Setup.** As discussed in Sec. 3, we perform the four stage training of the model. For pretraining (stages 1, 2 and 3) we use Cambrian2.5M dataset and perform instruction finetuning (stage 4) with the Cambrian10M dataset. We train our model on 8 Nvidia A100 GPUs with a batchsize of 16 per GPU. We adopt the pretrained LLaMA-1B and the 4-stage constrastively trained ConvNeXt as the base model for our attention-guided efficient VLM. We train our approach with two variants AGE-VLM which does not apply attention-guidance loss during finetuning (stage 4). This variant applies attention grounding only in the pretraining stage. The key advantage of this is that the self-attention layers are kept intact yielding a efficient way of introducing multimodal capability in the text-only LLM. In another scenario AGE-VLM-LM, we also apply guidance loss and train the model with the stage 4 as outlined in Sec. 3.

**Evaluation Metrics.** We evaluate our models on diverse benchmarks with each having a different metric to assess model performance. HallusionBench considers *aAcc*: the overall accuracy of all atomic questions, *qAcc*: the mean accuracy of unique questions as one question can be asked multiple times with different figures. A VLM correctly solved a unique question only if it succeeds in all <question, figure> pairs for this unique question. *fAcc*: the mean accuracy of all figures. One figure is associated with multiple questions, a VLM iscorrect on a figure only if it succeeds to solve all questions of this figure. CV-Bench consists of multiple choice questions, the models however, sometimes do not output the option even though they generate the correct answer. To account for this, we evaluate the accuracy by employing Qwen-L for evaluation. For OCRBench and RealWorldQA, we report the accuracy on the Scene-centric and the key information extraction tasks.

**Prior-art and Baseline.** We compare our approach against ConvLLaVA, MobileVLM-v2 and CA-baseline. ConvLLaVA also extracts vision features from ConvNeXt which are input to Vicuna-7B. MobileVLM-v2 with 1.7B parameters is based on CLIP-ViT with their MobileLLaMA, a downsized version of LLaMA. Both the models concatenate the vision tokens to the language tokens which are input to their respective LLMs optimized with the LM loss. We also include CA-baseline which has all the elements of our approach except for spatial distillation which attention guidance. That is in this variant the cross-attention layers and the self-attention layers are trained using only the LM loss.

**Quantitative Results.** In Tab. 3 we compare our approach to the prior-art and the baselines on efficient VLMs on different vision-centric benchmarks. We observe that on challenging datasets such as CV-bench our model outperforms prior work by a large margin. Similar improvements are

Table 4: **Attention visualization for different models.** Our method looks at the right regions based on the input image and the input text.

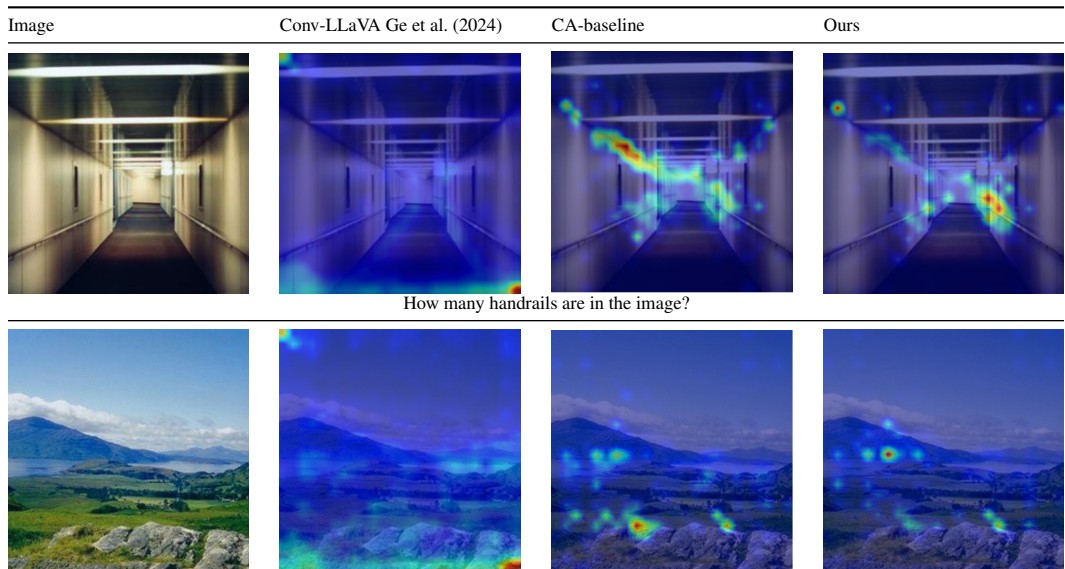

How many handrails are in the image?

Considering the relative positions of the river water and the stone in the image provided, where is the river water located with respect to the stone?

demonstrated on the OCRbench and the RealWorldQA datasets. This highlights the enhanced vision processing ability of our approach. Furthermore, we note that while our approach on HallusionBench yields better performance than ConvLLaVA, it is comparable to that of MobileVLM-v2. This can be attributed to fact that the attention signal from SAM cannot text information in mathematical charts or figures. Notably, our AGE-VLM variant trained in an efficient manner with self-attention layers intact (not trained with the grounding loss) consistently outperforms prior art with seamless integration of visual information with just 1.2B parameters.

**Qualitative Results.** In Tab. 3 we present the qualitative comparison of our AGE-VLM approach against Conv-LLaVA and MobileVLM-v2, the prior art of efficient VLMs on vision centric VQA task for challenging question and image pairs. Even though Conv-LLaVA answers incorrectly, it adheres to the instruction, answering with 0 or no. The responses generated by MobileVLM-v2 are not well grounded in the image as is evident from the explanation that follows the answer. For example,in column 3, MobileVLM-v2 incorrectly generates "soup" as the item in the image. In contrast, our approach not only follows the instruction but can also generate the response grounded in the image information. For a challenging case in column 2, our approach correctly localizes the location of apple in terms of the spatial relationship with other objects in the image and provides the correct response. We demonstrate the localization capabilities of our approach in Tab. 4. We visualize the attention weights of the first self-attention layer for Conv-LLaVA and the first cross-attention layer for the CA-baseline without attention guidance and our approach with attention guidance. As shown, given the image and the associated prompt, the Conv-LLaVA approach does not have any implicit grounding in the sefl-attention layer. The CA-baseline does have implicit grounding capacity but it incorrectly localizes the target visual concepts from the prompt. Our approach, on the other hand, localizes the correct regions, for example, handrails in row 1, and the river and stone in row 2

## 5  CONCLUSION

We introduced AGE-VLM, an efficient VLM designed to mitigate object hallucination. Our findings demonstrate that distilling knowledge from the SAM to guide attention mechanisms significantly enhances the visual grounding of VLMs. Extensive experiments show AGE-VLM achieves performance that is markedly improved or comparable to existing efficient VLMs on various vision-centric benchmarks. We believe this approach will stimulate further research into efficiently aligning vision and text modalities within the hidden states of pretrained LLMs with minimal overhead.

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

APPENDIX

**Limitations and Broader Impact.** While this paper has focused on the training recipe for distilling knowledge from SAM into vision-language models, our approach does not explore scaling of the distillation data or consider distilling optical flow or object tracking into the hidden states of VLMs. While our work addresses hallucination, it is far from perfect and can produce biased or factually incorrect content. With efficient VLMs as proposed in this work gaining traction, they will be widely accessable and should therefore be used with caution as their incorrect responses can cause physical harm such when using self-diagnosis with consulting medical experts.

We provide details on data collection for alignment guidance, additional details for training AGE-VLM, provide insights on further improvements with respect to the image data processing and include additional details on the evaluation benchmarks considered in the main paper. We also provide additional qualitative results.

## A  DATA FOR ALIGNMENT GUIDANCE

We leverage Grounded Segment Anything Model Ren et al. (2024) to obtain the masks of the target concepts to be focused on in the cross-attention layers. For text-based segmetation (referring expression segmentation) Grounded-SAM combines Florence-2 Xiao et al. (2024) and SAM ? to obtain the masks for the given text. Florence-2 takes a task instruction as input and generates results in the text form. Specifically for the referring expression segmentation, instruction "Ground the object which is most related to the text input" is provided. The segmentations are generated as polygons, with location tokens $(x_0, y_0, \ldots, x_n, y_n)$ representing the vertices of the polygon in clockwise order. The tokens and the image are provided to the SAM model to generate the target mask. With this pipeline, during the pre-training stages (1–3) we generate the target masks for 77K images and their associated captions in the Cambrian 2.5M dataset. Importantly, during fine-tuning, since the model takes image and a question prompt as input to generate the answer, we adhere to this framework and generate the segments based on the question for the given image. This instills in the model the ability to look at the right regions based on the question about the given image. For this phase, we utilize approximately 1% (150K samples) of the Cambrian10M instruction fine-tuning dataset.

## B  IMPLEMENTATION DETAILS

For any stage, we use the learning rate of $1e-5$ for all the modules including ConvNeXt, the projector, the cross-attention layers and the language model. We use Adam optimizer with the weight decay of 0.1, the warmup ratio of 0.03, $\beta_2$ is set to 0.95. Additionally, we train of each stage of a single epoch consistent with prior work on large vision-language models Ge et al. (2024).

## C  IMAGE PROCESSING AND ATTENTION

The input image to ConvNeXt is of size $768 \times 768$ yielding 576 tokens. We make an important observation the prior work Ge et al. (2024); Liu et al. (2023) zero-pad the images to resize them to target resolution. In our analysis we observe that for prior work without our attention guidance, the attention is focused on these padded regions. This might be an additional bottleneck for the vision-language models as they can easily ignore the vision features due to this inconsistency in the data. The impact of image-preprocessing techniques in large models needs further investigation and is an important direction for future work.

## D  EVALUATION BENCHMARKS

We specifically evaluate on vision-centric benchmarks which take into account the visual information for visual question answering, suitable for detecting hallucination in multimodal setting.

**HallusionBench Guan et al. (2024).**  The benchmark comprises 346 images paired with 1129 questions. The questions are framed in the yes/no format. The questions also ask about objects which

Figure 4: Comparison of the time to first token for cross-attention (black) to self attention (orange).

are not present in the image. This allows for targeted evaluation for our goal of attention guidance to integrate visual information emphasizing that the model "looks" at the image to perform the task.

**OCRBench Liu et al. (2024).**   It evaluates the ability of VLMs to accurately detect and read text in the images. In our model evaluation we focus on 475 images for the scene text centric VQA and key information extraction where images with text such as addresses, receipts, signs etc. are presented to the LMM and questions about the OCR content are asked.

**CV-Bench Tong et al. (2024a).**   This is a larger vision-centric benchmark containing 2638 manually-inspected examples. This contains image-question pairs to evaluate 2D (spatial relationships, counting) and 3D (depth order, relative distances) understanding of the VLM. For this it uses, ADE20k, COCO and OMNI3D dataset benchmarks.

**RealWorldQA x.ai (2024).**   This benchmark consists of 765 images, with a question and easily verifiable answer for each image. This dataset also contains questions about spatial understanding in images.

**POPE Li et al. (2023c).**   Similar to HallusionBench, POPE is also inspired to evaluate VLMs for object hallucination. This also contains yes/no format of the questions about the absence or presence of objects in the image.

## E   COMPARISON OF CROSS ATTENTION WITH SELF ATTENTION

The time to first token (TTFT) increases exponetially for the self-attention layers with the increase in number of visual tokens whereas for a model like AGE-VLM with interleaved cross-attention layers the TTFT is constant. This shows the need to re-evaluate the choice of models in VLMs that are efficient and yield vision centric capabilities.

## F   ADDITIONAL QUALITATIVE EXAMPLES

In Tab. 5 and 6, we present additional qualitative examples to show the performance of our model with attention guidance. Our model consistently performs better than Mobile-VLM V2 showing the advantages of our model in grounding its answers in the visual information. We support our results by visualzing the attention weights in Tab. 6 where our model is clearly able to look at the relevant regions for a given question.

| | | | |
|---|---|---|---|
| |  |  |  |
| Prompt | Is there a cup in the image? Please answer yes or no. | Is there a chair in the image? Please answer yes or no. | Is there a person in the image? Please answer yes or no. |
| Groundtruth | No | No | No |
| MobileVLM-v2 | Yes, there is a cup in the image. | Yes, there is a chair in the image. | Yes, there is a person in the image. |
| AGE-VLM | No, there is no cup in the image. The image primarily features fresh produce and does not include any cups or containers that could be used for drinking. The focus is on the display of fruits and vegetables, which are typically used for consumption rather than for drinking. | No, there is no chair in the image. The image is focused on the pizza and the person's hands, with no chairs or furniture visible in the frame. | No, there is no person in the image. The image solely features food items on a plate, which are the main subjects. There are no people present in the frame. |
| |  |  |  |
| Prompt | Is there a bicycle in the image? Please answer yes or no. | Is there a person in the image? Please answer yes or no. | Is there a person in the image? Please answer yes or no. |
| Groundtruth | no | no | no |
| MobileVLM-v2 | Yes, there is a bicycle in the image. | Yes, there is a person in the image. | Yes, there is a person in the image. |
| AGE-VLM | No, there is no bicycle in the image. The image is a natural scene with animals and vegetation, and it does not contain any man-made objects such as a bicycle. | No, there is no person in the image. The image solely features wildlife, specifically giraffes and zebras, in a natural setting. There are no human figures or indications of human presence in the scene. | There is no person visible in the image. |
| |  |  |  |
| Prompt | Is there a backpack in the image? Please answer yes or no. | Is there a chair in the image? Please answer yes or no. | Is there a bowl in the image? Please answer yes or no. |
| Groundtruth | yes | no | no |
| MobileVLM-v2 | No, there is no backpack in the image. The image features a black cat standing next to a bottle of wine. | Yes, there is a chair in the image. | Yes, there is a bowl in the image. |
| AGE-VLM | Yes, there is a backpack in the image. It is hanging on the wall to the right of the cat. | No, there is no chair in the image. The image shows a table with a pizza and a drink, and there is no chair visible in the frame. | No, there is no bowl in the image. The food is presented on a plate, which is a shallow dish typically used for serving food rather than a bowl. |

Table 5: Qualitative comparison of our approach against MobileVLM-v2 on vision-centric task.

| Image | Conv-LLaVA | CA-baseline | Ours |
|---|---|---|---|

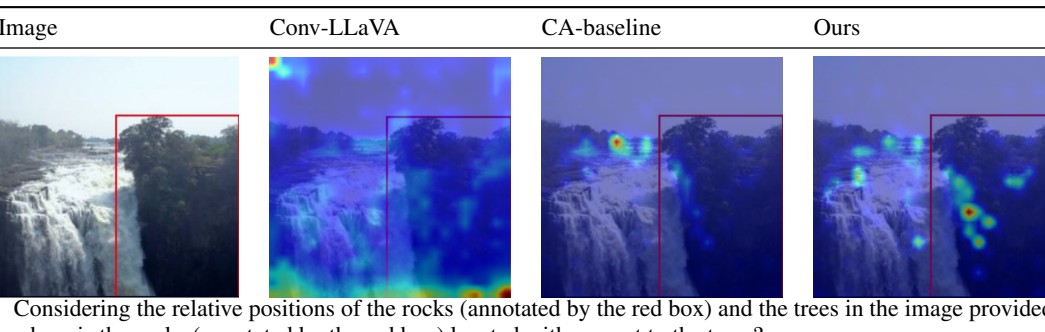

Considering the relative positions of the rocks (annotated by the red box) and the trees in the image provided, where is the rocks (annotated by the red box) located with respect to the trees?

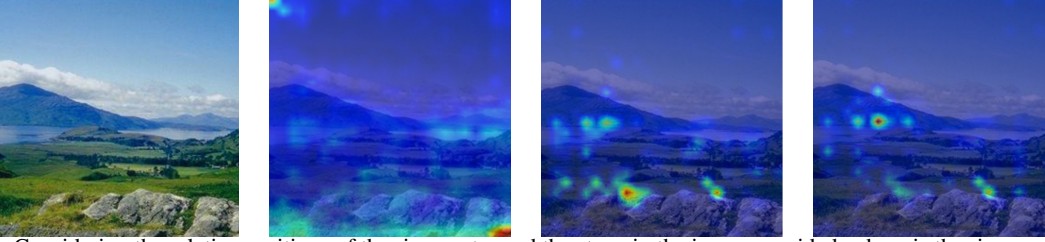

Considering the relative positions of the river water and the stone in the image provided, where is the river water located with respect to the stone?

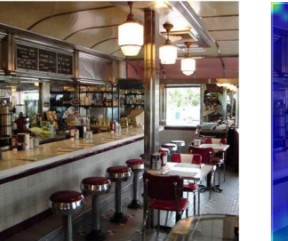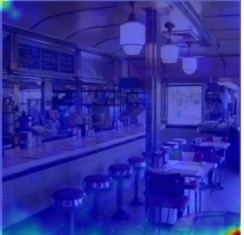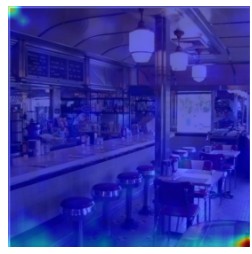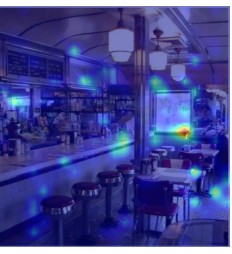

How many windows are in the image?

Table 6: **Attention visualization.** Our method looks at the right regions given the input image and the input text.

