# OpenReview forum: "Attention Guided Alignment in Efficient Vision-Language Models"
_ICLR.cc/2026/Conference — Submitted to ICLR 2026_

### Official Review · Reviewer_HTfK · 2025-10-24

**Soundness:** 2
**Presentation:** 3
**Contribution:** 2
**Rating:** 4
**Confidence:** 4

**Summary:**

This paper introduces AGE-VLM, an efficient VLM that reduces hallucination by interleaving cross-attention into a 1B LLM and supervising these attentions with SAM-derived segmentation masks via a Dice loss. The authors utilize ConvNext encoder and train the model with a four-stage-recipe, including alignment, vision adaptation, SAM-guided grounding, and instruction tuning. AGE-VLM achieves better or comparable results to baselines such as ConvLLaVA and obileVLM-v2.

**Strengths:**

1.	The paper is clearly-motivated and well-written.
2.	The idea of using attention alignment to improve the visual grounding capabilities of LVLM is very intuitive and makes a lot of sense.
3.	The use of SAM model for generating reference attention masks prevents the labeling process for GT attention, making the method efficient and scalable.

**Weaknesses:**

1.	While it is inspiring to improve the LVLM through attention alignment, the specific method used to align the attention (e.g., loss, optimization algorithm, architecture) is simple and standard.
2.	In Sec. 3.1, the authors examine visual–textual similarity and identify misalignment between visual and textual tokens. They then propose cross-attention as a superior architecture. However, many early VLMs already employed cross-attention to connect visual encoders with LLMs, whereas most recent models have shifted toward using simple adapters to project visual features into the textual representation space. This design appears to reverse that trend. Could the authors clarify the rationale for this choice?
3.	I am not sure whether SAM can give proper segmentation result when the prompt is complex or entails certain reasoning. And the SAM seems to fail in some OCR tasks. The authors should explain more about this.
4.	The performance of AGE-VLM-LM is inferior to AGE-VLM, which shows that attention alignment during visual instruction tuning will harm the performance. This is counterintuitive. Moreover, I wonder how well the attention alignment will be retained if there is no such regularization in the visual instruction tuning stage. The authors can explain more about this.
5.	In table 4, does ours refer to AGE-VLM or AGE-VLM-LM?  Besides the qualitative examples, are there any quantitative metrics to measure the alignment?
6.	The authors only include two baselines. More baselines should be considered such as those mentioned in the related work section (VILA, TinyLLaVA, Mini-Gemini).

**Questions:**

See the questions in the weaknesses.

---

> ### Author Response · Authors · 2025-11-28
> **Author Response to Reviewer HTfK**
>
> We thank the reviewer for their constructive feedback. We are glad that the reviewer finds our work well-motivated, intuitive, and our method efficient and scalable. We answer the reviewer comments below:
>
> **[W1] Simplicity of the approach.**
> Our approach is indeed simple to integrate in large language models and also can serve as an add-on on the VLMs. Our novelty lies in the formulation of the distillation technique (Eq. 1,2 & 3) where the cross-attention layers are interleaved in the self-attention layers of the LLM at various semantic depths to ensure that the models take into account the vision component when optimizing with next-token prediction in VLMs.
> We additionally introduce an efficient training strategy for integrating the cross-attention layers with minimum overhead on retraining or finetuning of the self-attention layers.
> We note that the LLMs have been trained on text-only data that is at least an order of magnitude larger than the available vision-text data used in VLM training. Thus, concatenation-based VLMs have weak vision priors and tend to ignore the vision component when performing vision-centric tasks.
> Our approach mitigates this and even with limited vison-text data, improves performance (e.g., over ConvLLAVA) across tasks in Tab.~2.
>
> **[W2] Design choice.**
> We found in the ConvLLaVA architecture that the simple projections and the self-attention architectures results require a large amount of vision tokens as the image resolution increases (576 tokens for resolution 1536).
> As the context window increases the image information has to be reinforced for example, by concatenating the image tokens again and this is limiting in efficient architectures where number of tokens is a bottleneck.
> To mitigate this, we revisit the cross-attention mechanism where each text-token attends to the vision tokens in the image. Thus, even with limited tokens (144 compared to 576), we achieve enhanced visual grounding compared to the purely projection based counterparts.
>
> **[W3] SAM and coarse segmentation masks.**
> Indeed, SAM cannot produce perfect segmentations. Notably, our approach, uses the segmentations of SAM as a weak-supervision signal during alignment stage to guide the model such that the text-tokens to pay more attention to the target regions. With our DICE loss formulation, however, we do not penalize the model for paying attention to other regions of the image. This guidance provides the model with the capability to look for target regions for a given text input; even for OCR tasks, where the segments are not explicitly available. The DICE loss can be interpreted as a regularizer on the attention maps, applied in addition to the next-token prediction loss, to enhance the vision capabilities of the VLMs.
>
> **[W4] AGE-VLM vs AGE-VLM-LM.**
> The performance of AGE-VLM-LM depends on the data available during the instruction fine-tuning phase. We note that with AGE-VLM-LM, we also fine-tune the self-attention layers which has been trained on closed text-only data. In a small LLM (LLAMA 1B), training with limited vision-text instruction tuning data compromises the language capabilities of the pre-trained backbone resulting in inferior performance as evident from the performance on OCR-bench which requires enhanced text understanding compared to other vision-centric benchmarks.
>
> **[W5] Table 4 and alignment metric.**
> In tab. 4, we refer to AGE-VLM as Ours.  We will fix this. We measure the alignment between the matching and non-matching pairs using cosine-similarity as shown in Fig.2. We do not explicitly measure the attention alignment as this is not the goal of the paper. The attention alignment can be measured by binarizing the attention map and measuring the overlap with the SAM masks. It is worth noting that our formulation does not penalize attention to other regions in the image and only rewards the target regions with the DICE loss. This ensures that the model takes into account the image context.
>
> **[W6] Baselines.**
> Our baselines, CA-baselines, ConvLLaVA show the benefits of our approach in improving vision-centric performance and reducing hallucination in efficient vision-language architectures. Baselines such as VILA, TinyLLaVA and Mini-Gemini utilize CLIP-VIT based vision encoders similar to the mobileVLM included in Tab.~2. These models also suffer from hallucinations and underperform on vision-centric benchmarks, for example, VILA (Llama-3-VILA1.5-8B) has an accuracy of 83.3 on POPE compared to our AGE-LM which has 87.34% accuracy on POPE. We will add this.
>
> We are happy to answer any further questions, thank you!

---

### Official Review · Reviewer_bU3e · 2025-10-28

**Soundness:** 2
**Presentation:** 3
**Contribution:** 3
**Rating:** 4
**Confidence:** 4

**Summary:**

The paper addresses the topic of hallucinations related to visual grounding caused by imperfect image-text token alignment in multi-modal models, demonstrates that image and text representations are not well correlated in architectures that concatenate text and visual tokens, proposes a cross-attention based architecture with a novel segmentation grounded loss and shows strong results on tasks that benefit from visual grounding.

**Strengths:**

1.	The paper tackles an important problem of object hallucinations in multimodal models and proposes a novel idea to guide attention to focus on relevant areas of the image using text grounded segmentation masks.
2.	The experimental setup is well formulated including multiple stages of pre-training and instruction fine tuning with the introduction of segmentation grounded loss in some stages. Specific focus is applied on maintaining language modeling performance.
3.	The results are well presented with evaluations covering different types of multimodal tasks like spatial reasoning, OCR, object detection among others. The qualitative examples are useful.

**Weaknesses:**

1.	Section 3.1 computes cosine similarity between final-layer hidden states at image-token vs text-token positions on matched and mismatched pairs, but it doesn’t clearly justify why hidden states are used instead of similarities in Q/K-space (Eg: cos(W_{q}h_{t}, W_{k}h_{v})
) which drive attention. It is unclear whether earlier/middle layers exhibit different alignment. How multiple tokens per modality are reduced is also not mentioned.
2.	Segmentation grounding loss is applied to 10% of the samples. Some ablations that show how this parameter affects learning would have been beneficial.
3.	Applying grounding loss to instruction following stage adversely affects performance on most of the benchmarks. This hasn’t been sufficiently addressed with observations of model behavior with and without this loss.
4.	Evaluations on language-only tasks and other general multimodal tasks including those which go beyond spatial grounding (like visual description) are missing.
5.	Minor typos and missing spaces.

**Questions:**

1.	Could the authors provide clarifications why the performance drops if grounding loss is applied in the instruction following stage?
2.	How was the fraction 10% arrived upon while applying grounding loss and are there any ablations to show how performance changes by varying this value?
3.	How do other general image understanding tasks which do not depend on object masks fare with this technique? Can this technique be scaled to state of the art general purpose multimodal models or do the loss functions need further modifications for different tasks?

---

> ### Author Response · Authors · 2025-11-28
> **Author Response to Reviewer bU3e**
>
> We thank the reviewer for their constructive comments and feedback. We are thankful to the reviewer for recognizing our approach as novel, well-formulated, and our results well presented.
> We address the reviewer concerns below:
>
> **[Q1] Performance in the instruction following stage.**
> When applying the grounding loss in the instruction following stage, the self-attention layers are also finetuned. The two losses are applied i.e., the DICE loss on the cross-attention layers and the next-token prediction loss to the self-attention layers. This finetuning is inefficient without access to the language only-data of the base LLAMA-1B model resulting in lower-performance of the language backbone.
>
> **[Q2] Data percentage in grounding loss**
> We note that we do not claim 10% data to be the optimal amount for training for SAM distillation. However, in our experiments, we found that 10% of the Cambrian data which corresponds to the amount of pre-training data in LLaVA stage-training is sufficient to achieve the attention alignment.  Scaling data could help, however, Cambrian2.5M also consists of the OCR datasets, the math datasets and text-only datasets to which SAM cannot be applied.
>
> **[Q3] Performance on tasks without SAM masks.**
> Since the cross-attention layers ensure that each text token is attending to the vision features, the approach generalizes to tasks such as OCR (as shown in Tab. 2). When performing SAM distillation, we encourage the model to look at the regions of interest. With the DICE loss formulation, we, however, do not penalize the attention on other regions. This serves as an additional guidance signal on top of the current next-token prediction loss and does not interfere with the general purpose capabilities of the large multimodal models.
>
> We are happy to answer any further questions. Thank you!

---

### Official Review · Reviewer_aEQA · 2025-10-30

**Soundness:** 3
**Presentation:** 2
**Contribution:** 2
**Rating:** 4
**Confidence:** 2

**Summary:**

The paper proposes AGE-VLM, a vision–language model that introduces interleaved cross-attention layers into a small LLaMA-1B backbone. The main idea is to guide the attention maps using segmentation masks distilled from the Segment Anything Model (SAM), improving spatial grounding and reducing hallucination. The method is trained in four stages that combine visual–text alignment, SAM-guided grounding, and instruction tuning. The numerical experiments suggest moderate but consistent improvements over prior efficient VLMs, with more clear visual localization and fewer hallucinated objects.

**Strengths:**

1. The paper discusses a relevant problem for efficient vision–language models.
2. The proposed SAM-guided cross-attention design is interpretable and integrates spatial grounding signals into a lightweight architecture without significant computational costs.

**Weaknesses:**

1. The SAM-guided cross-attention mechanism is relatively straightforward and appears as a simple extension of prior attention-alignment and grounding approaches.
2. The reported improvements do not look significant, and the method does not outperform the baseline by a considerable margin.
3. The role of each training stage and the specific contribution of the SAM supervision are not clearly disentangled, making it difficult to assess which component drives the observed gains.

**Questions:**

1. Can the authors clarify what distinguishes the proposed SAM-guided cross-attention from prior attention-guidance or grounding methods beyond using SAM masks for supervision?
2. Can the authors provide ablation results isolating the effect of the SAM guidance and each training stage to clarify which component contributes more to the numerical results?

---

> ### Author Response · Authors · 2025-11-28
> **Author Response to Reviewer aEQA**
>
> We thank the reviewer for their constructive feedback. We thank the reviewer for acknowledging our approach as a relevant and interpretable solution in efficient VLMs.
>
> **[Q1] Novelty compared to prior work with SAM as supervision.**
> In this work, we distill the SAM masks in the cross-attention layers which are interleaved into the self-attention layers of the LLM. Much of the prior work [a] applies Masks directly in the vision backbone, where the features corresponding to the context or other regions of the image are completely ignored. Our AGE-VLM approach, instead applies the masks only in the cross-attention layers guiding the model to pay "more" attention to the regions based on the content of the text or question. This design choice endows the model with capability to look at the target regions while also taking into account the context of the target when performing vision-language task in VLMs.
> [a] Efficient Vision-Language Pre-training by Cluster Masking. CVPR 2024.
>
>
> **[Q2] Isolate SAM guidance and other training stages.**
> We isolate the effect of SAM guidance, in our CA baseline (without SAM guidance; [Tab. 2; row 1]) where the cross-attention is applied equally across all the vision-features in an image. The performance of the model with Convolutional backbone and Llama series of models, without SAM distillation and cross-attention, is given by ConvLlaVA [Tab. 2; row 2]. As shown in Table 2, the ConvLLaVA approach, despite its stronger language model, underperforms the CA-baseline without cross-attention on different benchmarks, especially on the vision-centric tasks e.g., POPE. This highlights the limitation of concatenation-based VLMs in understanding the associations between the vision and language modalities. With cross-attention layers, each text token has the capacity to attend to vision-tokens, thereby, reinforcing the utilization of vision features when performing vision-grounded tasks.
>
> We are happy to answer any further questions.

---

### Official Review · Reviewer_fCwf · 2025-11-01

**Soundness:** 3
**Presentation:** 3
**Contribution:** 3
**Rating:** 4
**Confidence:** 4

**Summary:**

The paper targets object hallucination and weak visual grounding in efficient VLMs that use concatenation of visual/text tokens. The authors diagnose the issue via cosine-similarity analyses showing that hidden states for matching and non-matching image–text pairs overlap substantially, indicating poor multimodal alignment. They propose AGE-VLM, which (i) interleaves cross-attention layers inside a small LLM (LLaMA-1B) and (ii) guides those cross-attention maps using masks distilled from (Grounded) SAM during pretraining/fine-tuning (“attention-guidance loss”), so the model “looks” at the right regions.

**Strengths:**

- The similarity-distribution study nicely evidences why concatenation architectures blur matching vs. non-matching pairs—useful and reproducible diagnostic
- Distilling SAM masks into cross-attention (not the vision backbone) is conceptually clean and data-efficient; the dice loss formulation is appropriate for sparse regions
- Interleaving cross-attention in a 1B LLM while mostly freezing self-attention preserves language priors and keeps training economical; the staged plan is easy to adopt
- Quantitative tables plus qualitative heatmaps support the claim that AGE-VLM reduces hallucination and improves localization

**Weaknesses:**

- Evaluation protocol introduces an external judge: For CV-Bench, accuracy is computed by Qwen-L because models sometimes omit option letters, this can inject evaluator bias and hides raw option-selection accuracy
- Heavy reliance on SAM/Grounded-SAM: Performance hinges on third-party segmentation quality and prompt engineering; generalization when masks are imperfect/noisy is not stress-tested (authors acknowledge broader-impact limits)
- There is a CA-baseline (cross-attention without guidance), but more granular ablations (e.g., how many cross-attn layers, which tokens supervised, mask quality/noise curves) are not detailed in the main text.
- Narrow model/backbone space: Results center on ConvNeXt+LLaMA-1B; comparisons to stronger small-VLM baselines or ViT backbones at similar token budgets are limited, despite token-efficiency claims

**Questions:**

- The overlap in hidden-state cosine similarities is compelling, but does improving that metric necessarily reduce hallucination—or does the SAM-guided training improve both due to an external supervision signal?
- Using Grounded-SAM (Florence-2 + SAM) to create supervision during both pretraining and instruction fine-tuning could introduce benchmark leakage if those systems saw similar data distributions.More detail on dedup/filtering and robustness to mask errors would help

---

> ### Author Response · Authors · 2025-11-28
> **Author Response to Reviewer fCwf**
>
> We thank the reviewer for their constructive feedback and comments. We are glad that the reviewer finds our evidence on similarity distribution useful and reproducible diagnostic and our framework data efficient and clean.
> **[W1] Evaluation protocol introduces an external judge.**
> We observe that models have a tendency to output the answer itself instead of the options at all time. So as not to penalize the models for not answering the options directly, and the associated answer, thereby testing its understanding and accurate modeling capabilities.
>
>
> **[W2] Heavy reliance on SAM/Grounded-SAM, robustness to mask errors.**
> Indeed, SAM cannot produce perfect segmentations. Notably, our approach, uses the segmentations of SAM as a weak-supervision signal during alignment stage to guide the model such that the text-tokens to pay more attention to the target regions. With our DICE loss formulation, however, we do not penalize the model for paying attention to other regions of the image. This guidance provides the model with the capability to look for target regions for a given text input. The DICE loss can be interpreted as a regularizer on the attention maps, applied in addition to the next-token prediction loss, to enhance the vision capabilities of the VLMs.
>
>
> **[W3] There is a CA-baseline and granularity of cross-attention.**
>  The interleaved cross-attention layers in VLMs have been applies at levels of varying sematic depth (at index  2, 7, 12, and 17) for 16 decoder layers in LLaMA-1B. This is consistent with prior work such as Flamingo and provides a balance between efficiency of the underlying small LLM and expressiveness (of an integrated multimodal module with cross-attention).
>
>
> **[W4]  Comparisons to stronger small-VLM baselines or ViT backbones.**
> --- Baselines such as VILA, TinyLLaVA and Mini-Gemini utilize CLIP-VIT based vision encoders similar to the mobileVLM included in Tab.~2. These models also suffer from hallucinations and underperform on vision-centric benchmarks, for example, VILA (Llama-3-VILA1.5-8B) has an accuracy of 83.3 on POPE compared to our AGE-LM which has 87.34% accuracy on POPE. We will add this.
>
> **[Q1] Cosine-similarity and hallucination.**
>  In our work, we do not optimize the cosine similarity metric and use a DICE formulation in Eq. 3 for distilling SAM to guide the model's attention to target regions. The histogram in Fig. 2 supports our hypothesis and the widely noted observation that VLMs tend to ignore visual information and have a strong bias for the language priors. This is also observed in the attention maps where the attention the attention weights do not necessarily look at the right regions (Tab. 4). To address this, we make the attention explicit by distilling SAM features for which we apply the DICE loss and do not employ any cosine similarity metric during training.
>
> **[Q2] Data distributions and benchmark leakage.**
> We provide the details on alignment data used in Appendix A. To ensure that we perform training on the same vision-text data as prior work, we use the data from Cambrian, curated to minimize benchmark leakage. Since we apply SAM distillation through DICE loss which rewards the attention on right regions without penalizing the attention on other regions by the model, our framework is robust to coarse masks. Moreover, the DICE loss is complementary to the next-token prediction loss.
> In our training, we filter out the vision-text data for which SAM does not yield any mask, for example, OCR subset. We will clarify the details in the paper.
>
> We are happy to answer any further questions, Thank you!

---

### Meta-Review · Area_Chair_fmLf · 2026-01-07

**Summary:**

Several reviewers noted that the method’s heavy reliance on SAM/Grounded-SAM may be brittle under noisy or imperfect masks. They also raised concerns about the evaluation protocol—particularly the use of an external LLM as a judge—and the possibility of data leakage. In addition, reviewers felt the ablation studies and baseline comparisons were insufficient to cleanly attribute gains or substantiate the paper’s token-efficiency claims.

**Reviewer Concerns:**

I think the authors address the concerns about the experimental and method clarification such as "cosine-similarity diagnostic vs. hallucination causality", the authors also tried to justify the usage of external LLM as a judge but I am still not convinced. Besides, the robustness issue of using SAM/Grounded SAM is an outstanding concern too. The new updated experiments still look coarse and not convincing.

**Reviewer Scores:**

I do not think the reviewers will raise the scores.

---

### Decision · Program_Chairs · 2026-01-26

Reject